# Life Cycle Assessment to Ensure Sustainability of Circular Business Models in Manufacturing

**Marit Moe Bjørnbet [1],\* and Sigurd Sagen Vildåsen [2]**

[1] Department of Industrial Economics and Technology Management, Norwegian University of Science and Technology, 7491 Trondheim, Norway

[2] Department of Industrial Ecosystems, SINTEF Manufacturing, 2830 Raufoss, Norway; sigurd.vildaasen@sintef.no

\* Correspondence: marit.moe.bjornbet@ntnu.no

**Abstract:** Circular business models (CBMs) represent a path for coordinating circular economy (CE) efforts. Life-cycle assessment (LCA) is a tool for quantifying environmental effects of a product or a service and can therefore evaluate the environmental sustainability of CBMs. This paper explores whether LCA can act as an enabler for manufacturing companies who want to implement a CBM. Following a case-study approach, we draw on the experiences of a specific manufacturing company during the time period 2014–2021. The paper presents key lessons on the interaction between LCA and CE. The study finds that LCA—by providing quantified results on the environmental impacts of circular strategies—limits the risk of problem shifting and challenges the normative rule of closing the loop by including a set of multiple impact categories. LCA offers a common platform and encourages communication with stakeholders. These characteristics make LCA a well-suited tool for CBM development. However, the holistic perspective on environmental problems that LCA provides can also complicate CE's clear message of 'closing the loop'. Lastly, LCA is a tool for environmental evaluation, and with the main emphasis of CE also on environmental issues, there is high risk of neglecting social and economic aspects of sustainable development.

**Keywords:** circular economy; sustainable development; case study; life-cycle assessment; circular business model; circular manufacturing

## 1. Introduction

Manufacturing of products is important for providing acceptable living standards for people while at the same time contributing to environmental problems. The role of production and consumption in this is acknowledged by the United Nations through the agenda for sustainable development [1] and more specifically through SDG no. 12 to ensure sustainable consumption and production patterns. Whilst awareness of the unsustainable production and consumption patterns of humanity has grown, circular economy (CE) as a route to change this development has gained traction. CE is about decoupling economic growth from environmental degradation [2], and the concept is linked to the SDGs through target 12.5 to substantially reduce waste generation through prevention, reduction, recycling, and reuse. Because CE is a means to increase circularity by shifting focus from downstream processes (waste collection and processing) to fundamental changes in upstream processes of production and consumption, all actors across the product chain are affected [3]. Manufacturing companies play an important role in CE, especially in their possibility to affect the use stage of products [4], and thus nudging activities higher up in the waste hierarchy. However, most manufacturing SMEs are still on a low stage of CE implementation [5]. For companies, a suggested route for CE implementation is through circular business models (CBMs) [6–10]. CBMs offer a promising path for CE implementation, but there is still limited support in the transition phase from linear BMs to CBMs [11].

The efforts to achieve the replacement of the EOL concept [12] represent a set of CE strategies that can be employed and are often presented by a number of Rs (e.g., reduce, reuse, recycle and recover) also known as the waste hierarchy [13]. The Rs (or CE strategies) can be categorized into three groups: short loops (users' choices), medium long loops (product upgrade) and long loops (downcycling) [14]. From a CE perspective, the shortest loops provide less strain on natural resources and are preferred over the longer loops. Circular business models (CBMs) provide a way to target CE efforts. However, reviews of CBM literature show that multiple understandings of what CBMs are exists [11,15]—from some defining CBMs by applying economic value creation as the core construct without describing the environmental dimension [16], to others [17] emphasizing that environmental benefits of CBMs should be made explicit and verifiable by stakeholders. To maintain the link to sustainability, it is necessary to select a holistic CBM definition that combines the environmental and economics dimension to consider the societal system in which business operations take part. A review [18] includes the following definition (p. 12):

*A circular business model is how a company creates, captures, and delivers value with the value creation logic designed to improve resource efficiency through contributing to extending useful life of products and parts (e.g., through long-life design, repair, and remanufacturing) and closing material loops.*

The definition emphasizes that the economic value creation logic should be designed to improve the environmental dimension. Moreover, it explicates the environmental dimension by means of two strategies: life-time extensions and closed material loops. These two CE strategies are well-known in the CBM literature [19]. Enhanced resource efficiency throughout a product life cycle requires not focusing solely on one single but exploring combinations of CE strategies [20]. In other words, CBMs can consist of one or several CE strategies. Moreover, rather than a concise change, CBM development tends to occur as an iterative, emergent process [7] and through learning and experimentation [21]. Thus, CBM transformation can be divided into three phases: (1) identifying problems and solutions, (2) evaluate solutions and (3) implement solutions [22].

The role of involving stakeholders in CBMs is highlighted by several authors [10,21,23,24]. Stakeholders can be identified in a broad sense as any "group or individual who is affected by or can affect the achievement of an organization's objectives" [25] (p. 46). Because the aim of CE is to close resource loops, implementation success requires efforts outside the company borders. A CBM aiming to extending useful life of products requires a life-cycle perspective involving stakeholders [26]. On the other hand, progressing towards a new CBM might cause tensions between the initiating company and [7]. This highlights the importance of transparency and communication between and towards stakeholders [27,28].

Despite the explicit connection between CE and sustainable development made by the SDGs (through target 12.5), the link should not be taken for granted. A review of CE definitions finds that most lack a reference to sustainability [12]. Evaluating the effects of CE efforts such as CBMs is key for securing their contribution to sustainability [29]. However, though many companies track and store CE relevant information for reporting, data are not to a large extent analyzed or used in decision-making processes [5]. Measuring the effects of business model changes for CE progress quantitatively is difficult [7].

Life-cycle assessment (LCA) is a methodology for quantifying environmental impact of products, processes, and services. The use of LCA is standardized trough the international standards ISO 14040 and 14044 [30,31]. LCA is applied for learning (improvement possibilities, environmental performance indicators), communication (eco-labels, environmental product declarations, benchmarking) and for decision making (design and development of product and processes, purchasing, development of policies and regulations) [32]. LCA follows four stages [31]. First, the goal and scope definition, where the purpose assessment is stated, and the plans are made accordingly. This is the stage where the functional unit and system boundaries is defined. Second is the life cycle inventory analysis (LCI), where data for all relevant inputs and outputs (e.g., materials, transport, energy use, waste, and emissions) are collected. The third stage is the life-cycle impact assessment (LCIA) where

the environmental effects are evaluated based on the inventory analysis, and lastly, the interpretation stage where the results are interpreted and discussed. In the somewhat ambiguous field of sustainability reporting, LCA offers a way of communicating environmental impact of products and services, in facts and numbers. Through the method's diverse impact categories, LCA also provides insights into possible trade-offs and potential risk of problem shifting, i.e., solving an environmental problem by creating a new one.

As CBMs require efforts from beyond the company's borders, and so does LCA through its life cycle perspective. Data is collected from the entire life cycle of the product (or service) under assessment, often involving stakeholders such as suppliers, users and end-of-life (EOL) firms. LCA is presented as a suitable tool to evaluate the environmental performance of CE efforts [33,34]. However, applying LCA on CE practices entails methodological challenges and requires new LCA standards implementing gains from material cascading in CE solutions [35]. Exploring the role of LCA in development of CBMs, [36] argues that a full LCA can be applicable in the later phases of development, whereas streamlined LCA is more relevant earlier (when parameters are more uncertain and fuzzier). One approach proposed by [22] is applying LCA in the later phases of CBM development (after design of solutions) to choose between different strategies and evaluate the performance. Both [17] and [37] argue that a reference system is needed to assess the environmental impacts of new BMs, and that this is causing challenges because many CBMs represent new solutions. For systems that do not exist yet, but are being developed, defining suitable system boundaries and obtaining data is challenging. Identifying functional-unit and setting-system boundaries are challenges that occur when evaluating PSS (product/service-systems) with LCA [37]. A systematic review [38] shows that LCA is the assessment method most frequently used for CE performance assessment. LCA can be used to evaluate and compare circular strategies to existing strategies (i.e., traditional BMs) [26,35,39]; on the other hand, [40] suggests using assessments as a point of departure for circular strategies.

LCA offers a route for evaluating CE initiatives' progress towards environmental sustainability [33,34]. The need for more support on CE implementation and evaluation of its contribution to sustainable development makes it interesting to explore how LCA can help to direct CBM development and safeguard environmental sustainability. The goal of this paper is therefore to explore if LCA can act as an enabler for manufacturing companies who wants to implement a circular business model. Through a longitudinal exploratory-case-study approach, we follow a Norwegian manufacturing company to shed light on how their work with LCA and CBM development is connected. Relying on the experiences from this study, the findings are connected to the existing literature to gain in-depth insight into the interplay between LCA and CBM.

This paper starts with providing a theoretical background of LCA and CBM (Section 1). Further, a presentation of the research method and the case company is given (Section 2), before results (Section 3), and discussion is presented (Section 4). Last are some final conclusions, and suggestions for further research is provided (Section 5).

## 2. Materials and Methods

This paper describes an exploratory longitudinal case study. Case studies enable theory building in close connection to empirical reality [41] and are a suitable research method for investigation of a contemporary phenomenon, with multiple sources of evidence, with more variables of interest than data points, where existing theory is used to guide analysis of data. Moreover, a single case study provides the opportunity for a deeper analysis of a research phenomenon. An exploratory approach is chosen for this case study due to the immature nature of the topic [42]. An exploratory longitudinal case study, of the type performed in this case, is time consuming and difficult to replicate. However, according to [43], longitudinal case studies come with the opportunity to follow a 'developmental course of interest' and see how conditions change over time, and therefore they offer a rationale for a single case study approach. Therefore, through an exploratory longitudinal case study

approach, the company's journey (and LCAs role in it) from a traditional BM towards a CBM is investigated. The next sections describe the case company, the rationale behind the case selection, and further how data is collected through interaction, communication, and workshops.

The case company is a Norwegian manufacturer of composite liquefied petroleum gas (LPG) cylinders that are distributed worldwide. The company was established in 1996 and started selling products in 2000. The case product, a composite LPG cylinder, is a complex, high-tech reusable product for both consumer and business markets. The life cycle of the product is intricate due to their worldwide distribution it is used for different purposes in different markets. Product lifetime is estimated to be 30 years, with service and maintenance intervals of approximately 10 years. Figure 1 shows the main life-cycle stages of the case company's product. The case company is a B2B actor; e.g., they sell their product to a distributor who is responsible for maintenance and repair. The distributor further sells the product to a retailer, who rents out the product to the end user. The case company is a suitable case for an exploratory longitudinal case study because of their engagement in environmental activities, their worldwide product markets, and the associated complexity in the value chain. The researcher's long relation with the case company also provided a unique opportunity to collect and analyze data over a longer period, and thus explore development over time. Further, the case company's role in an industry with challenges regarding EOL, the core construct of CE, also makes the company a relevant point of departure for exploring the goal of this paper.

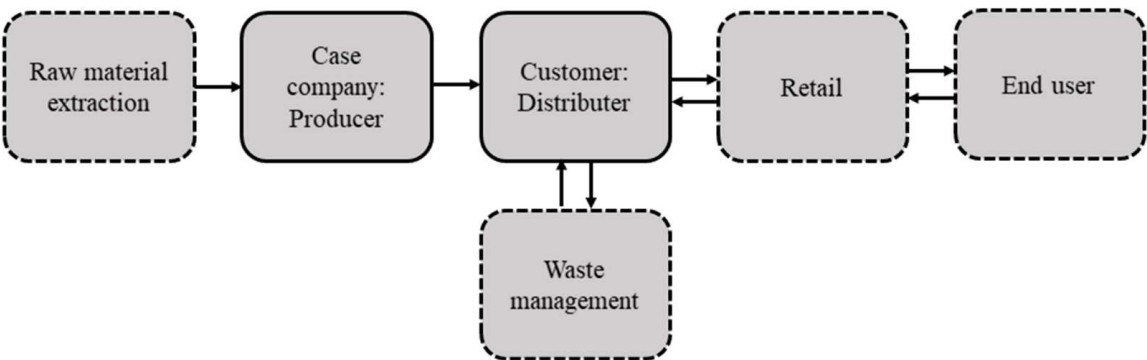

**Figure 1.** Life-cycle stages for the case company's product. Data is collected from case companies and customers (solid lines); other actors have dashed lines. Arrows show the physical transport of the product during its lifetime.

A good case study comprises multiple sources of evidence [43]. This study relied on several data sources such as documentation (emails, meeting notes, reports), direct observations of meetings and workshops as well as participant observations (in which the researcher is an active participant) gathered during the time 2014–2021. Figure 2 shows the period for this longitudinal case study. It started with a research project in 2009—Sustainable Manufacturing of Light Weight Solutions (KBM SUM)—where a screening LCA was performed with the goal of evaluating the environmental impact of different EOL options for the product. The researchers were not an active or observative part of this research project; the report from the LCA-screening was analyzed as a secondary data source. From 2014 to 2019, the company was part of a competence project called Sustainable innovation and shared value creation in the Norwegian industry (KPN SISVI). The researcher's involvement with the case company started in this competence project in 2014; consequently the data collection with the researchers as active participants started, i.e., primary data sources. The data sources from this research project ranged from meeting notes and observations, field trip to email communication. Further, a comparative LCA was performed during the project period, and the report is one of the data sources analyzed in this case study.

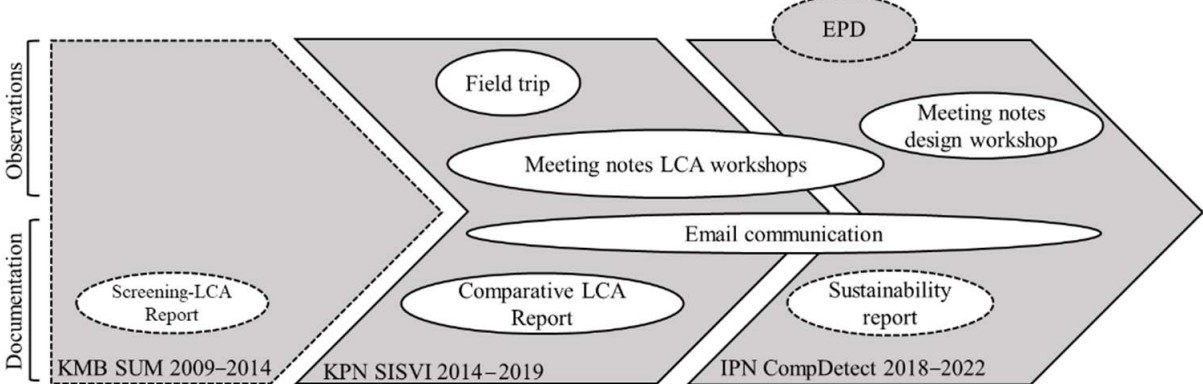

**Figure 2.** Timeline showing the data sources and how they relate to the company's engagement in research projects (arrows) from 2009 to 2022. White filled circles represent data sources, primary data sources are solid circles and secondary dashed circles. The circle filled with grey color represent an activity performed by the case company outside the research projects, but still relevant for the scope of this paper.

Following the competence project, the case company (along with other industry partners) was granted an innovation project named Smart Sustainable Composite Products (IPN CompDetect) in 2018. The project, with the aim to create CBMs by addressing the EOL stage of products and strategies for extending useful life of product and parts, is not yet completed. Data sources from this period ranges from documentation such as a sustainability report published by the case company themselves in 2019, email correspondence, as well as observations (active and passive) of workshops. Further, the company initiated an environmental product declaration (EPD) in 2020.

Data analysis in a case study involves examining, categorizing, and testing data to produce empirically based findings [43]. This case study started out by an inductive research approach, with specific observations collected over time; thus, the analysis strategy was to "work the data from the ground up" (ibid.). As the analysis progressed, the researchers shifted towards a more deductive analysis strategy of structuring the data through a theoretical framework to investigate the fit with findings in the existing literature on the topic.

## 3. Results

The goal of this paper was to explore the use of LCA in the development process of a CBM. This section first presents the findings of the exploratory longitudinal case study. The case company was engaged in several LCA activities during the time of this study: a 'screening LCA' with the purpose to evaluate EOL options, a 'comparative LCA' to compare the composite LPG cylinder with a steel LPG cylinder, a carbon footprint analysis, and lastly an EPD. The following sections describe the experiences from the LCA activities in the longitudinal exploratory case study that are relevant for CBM development and are framed and structured according to the four stages of LCA. CBMs can consist of one or several CE strategies. The two strategies presented by [18] offer a good fit with the case company's focus in the period of the case study, so the focus of this paper is therefore on the two CE strategies 'extending useful life of products and parts' and 'closing of material loops'.

### 3.1. The Role of Goal and Scope Definition

The goal and scope definition of an LCA involves specifying a functional unit that is consistent with the intended purpose of the LCA and determining the system boundaries [31]. The purpose of each LCA was motivated by the company's strategic needs, which were multiple and varied over time. Neither of the LCA's were designed to evaluate CE strategies directly, but they originated from reasons such as competitive market reasons, customer inquiries to motivation to contribute to a more sustainable and circular product.

For each of the LCAs, the functional unit and system boundaries were set according to the purpose, which further affected the results. A minor reduction in the use of material in production provided little change in environmental impact per functional unit in the comparative LCA (with functional unit liter delivered gas to end user) but nevertheless was deemed fruitful contribution to more circularity through the contribution towards improved resource efficiency for the case company. This complicated the communication LCA results and required flexibility when the LCA results was presented and discussed with the case company. In contrast to LCA, which is usually product oriented, CBM has a wider scope, i.e., how a company creates, captures, and delivers value [18], and thus efforts that contribute to less resource use are appraised.

Initial work with defining scope and system boundaries for LCA revealed that the case product's EOL procedures were very region specific. Providing data from all markets was unrealistic and too time consuming. It would also have relied highly on customer relations and their ability to acquire the needed data. The LCA that involved use and EOL therefore used a scenario approach. This meant that one customer and market was targeted for the assessment. The scenario was chosen based on data accessibility (i.e., how available and certain were the data) for the LCA, and not to represent the average market conditions. Using such an approach limited the generalizability of the results, and further the usefulness for CBM development. For the LCA to be more valuable in CBM development and maximize and ensure the impact of the CE efforts, a broader part of the markets, i.e., multiple scenarios, could have been beneficial. This encouraged the case company to initiate parameterization of data to be used in calculation tools, with the goal to provide environmental impact information for specific use/EOL cases for their customers.

### 3.2. The Role of Life Cycle Inventory Analysis

In the second step of LCA (LCI), data for all input and output is collected [31]. Data collection is a crucial and often time-consuming step of LCA because it provides the foundation for the impact assessment. For the LCAs performed in this case study, data collection for production of the case product was a relatively straight forward task because the case company itself had access to the data needed. Data from suppliers required efforts from the case company but was achievable to be held off. The uncertainty in data grew when addressing the use and EOL stage. This was partly because the case company was a company in the B2B segment, not directly in contact with their end users, but also because their products had multiple use areas, markets, and associated regulations and EOL behavior. Data collection required involvement from stakeholders further up/down in the value chain, and this triggered efforts to involve customers (not end users). This resulted in a field trip to provide knowledge on use practices, repair, and maintenance as well as EOL handling to improve the inventory analysis part of the LCA. The insights turned out to be valuable beyond the LCA: the involvement of stakeholders (waste companies, customers, suppliers) and relationship building was also important for the CBM development process. Just as LCA requires insights from outside the company borders, CE efforts cannot succeed in isolation from the surrounding system but require collaboration and communication with stakeholders [10,23,24].

For the use stage, estimates of lifetime was essential. Due to the long lifetime of the case product, there was no accurate estimates of product lifetime. The production started in 2000, so most first-generation products were still on the market. Regulations and requirements varied from region to region and market to market, from required discarding after x no. of years in some markets, to the regulated test intervals in others. The lifetime estimate was therefore developed in collaboration with a customer with good insight and experience with the case product. There were, however, uncertainties in these estimates that also influenced the robustness of the results, their generalizability, as well as their applicability in decision making. The attempts to increase knowledge on the lifetime of the case product were initiated as a part of the inventory analysis and were experienced as challenging due to the high level of uncertainty. However, the process led to increased

awareness on how the lifetime of each case product affected the LCA results and motivation to explore the reasons for early discarding of some products. Further, the case company initiated new research projects on models to predict product lifetime.

### 3.3. The Role of Life Cycle Impact Assessment

The third stage of LCA, the life-cycle impact assessment (LCIA), is where the environmental effects are calculated [31]. The scope of this paper was not the impact assessment results from each of the LCA studies but how the results were utilized for CBM development. The LCA results from the comparative LCA showed that the production stage was responsible for a large share of impacts for all selected environmental impact categories. Most of this could be traced back to material use. Based on this, the company initiated sourcing strategies, i.e., dialogue and collaboration with supplies to reduce the impacts from materials. For global warming potential, the use stage also contributed significantly. This was mainly due to transportation in the use stage, and electricity used for re-test and re-furbishment. The LCA results showed that the EOL of the case product did not contribute much to the overall environmental impacts. This was influenced by the way the functional unit was set, and how the goal and scope of the LCA was designed. Further, the results of the first LCA screening in 2012 (targeting EOL options for the case product) showed that incineration (with energy recovery) was a favorable EOL option when compared to recycling in cement kilns and landfills. The small contributions from EOL in the comparative LCA sparked a discussion on the importance of focusing on this stage in the product's life cycle to reduce environmental impact. The case company is currently receiving feedback from customers requesting recommendations for EOL handling. Further, they were confronted on the (lack of) recyclability of their product. Although the results revealed small contributions from EOL, the process of working with the LCA had, for the case company, provided new insight into the EOL treatment of the product. This, along with customer feedback, triggered an interest to further explore possible waste scenarios for the product. The case company wanted to be able to provide customers recommendations for EOL treatment as well as to limit the case products negative impact on the environment. This focus resulted in targeted efforts to develop an innovation project in collaboration with other industrial composite partners. The scope of this innovation project initially excluded incineration as an EOL strategy despite it being deemed the most favorable option in the LCA screening. Incineration was left out for the benefit of other 'more circular' strategies such as recycling, refurbishment, and prolonging lifetime.

### 3.4. The Role of Interpretation

The fourth stage of LCA—interpretation—is where the results are explained and interpreted [31]. All of the LCA activities in this study were communicated by LCA practitioners to case company representatives, through reports and presentations. One of the benefits of LCA as a scientific method is the broad set of impact categories applied. This ensures that a solution suggested to cope with one environmental problem is not shifted to another environmental issue (problem shifting). However, during the communication of results in this case study, the many impact categories also raised the question, "what is best?", which is impossible to answer. One solution improved performance in one environmental impact category but increased negative impacts in another. A recurring theme throughout this case study was the tension between the case company's wish for the one right answer, and the complexity LCA as a method offers. LCA, suggests through its quantitative nature that one correct numerical answer exists. This was intriguing for the case company looking for clear results for market communication. This led to potential pitfalls and challenges when communicating results. This study found that, for the sake of measuring and quantifying, LCA can introduce more complexity into CE efforts and thus hamper companies' attempts to 'do the right thing'.

## 4. Discussion

### 4.1. Implications for Circular Business Model Development

This section links experiences from the case study to the existing literature on CBM development and discusses the role of LCA in safeguarding the sustainability of CE efforts. Lifetime extensions and closing of material loops are the two strategies suggested by [18] for companies to build their value creation logic on (i.e., creating, capturing, and delivering value in a resource efficient manner). For the case company, it was critical to understand how the case product affected the environment, also compared to competing products. Competitive reasoning was also a contributing aspect for the case companies' motivation. In the sustainability report, the response to the LCA results showing relatively higher environmental impacts from the EOL stage of their product, compared to steel was "several ongoing strategic actions and research projects have been put in motion in order to find sustainable technologies for EOL treatment" [44]. The relative share of impacts from EOL was low, but nevertheless, this was the life-cycle stage where the case product performed worst relative to the competing product, and therefore it was important for the case company to improve. LCA does not incorporate economic sustainability, a critical shortcoming considering manufacturing companies' underlying goal of economic benefits [9]. However, for the case company, there was a connection between environmental impact reduction and economic savings both through their efforts to reduce impact from materials as well as in energy-saving measures in production. Table 1 shows the findings from the case study for the four stages of LCA and how the stages affect CBM development for the two circular strategies: lifetime extensions and losing of material loops.

**Table 1.** Case study findings structured through the four stages of LCA.

| Stages in LCA | Implications for CBM Development/Experiences from Case Study | |
|---|---|---|
| | **Lifetime Extensions** | **Closing of Material Loops** |
| 1: Goal and scope definition | Initiating LCA led to increased awareness on the potential benefit of measures to increase lifetime of case product and prevent discarding of useful products. | No one-size-fits-all solution for CE strategy for closing of material loops existed due to a wide range of existing EOL practices and market-dependent barriers for new EOL technologies. To be of greater value for CBM development, LCA should have included multiple scenarios for EOL. |
| | In LCA, the goal, scope and functional unit directs the results and therefore should have been set to provide a better fit with CBM development. The LCA was product oriented; a CBM is not (only) product oriented but often has a wider, organizational scope. This resulted in LCA results not being directly applicable for CE evaluation | |
| 2: Inventory analysis | Lifetime predictions for the case product was uncertain, and data collection was difficult. The use stage was a challenging stage to obtain data from, and at the same time it is the essence of the CE strategy to extend useful life of products and parts. | Data collection on new EOL technologies for the case product was challenging and involved uncertainty. The EOL stage was a challenging stage to obtain data from, and at the same time it is the essence of the CE strategy to close material loops. |
| | This stage of LCA involved data collection from all parts of the product's life cycle; this fostered communication and collaboration with stakeholders useful for CBM development. | |
| 3: Impact assessment | LCA results motivated implementation of strategies for lifetime extensions. Further, LCA provided quantified results on improvements that extended the lifetime of products and parts. | LCA provided quantifiable results for different EOL options. However, there was not always a perfect fit between impact categories and the normative rules of the CE concept. |
| | Avoids problem shifting, but also complicated evaluation of CE strategies. | |
| 4: Interpretation | LCA contributed to awareness raising on the role of lifetime extension strategies. The case product had a long, uncertain, and variable lifetime, which resulted in limitations in possible use of LCA results. | LCA contributed with awareness raising on the role of closing of material loops. For the case product, with large differences in markets, LCA results from EOL were either scenario specific or difficult to use for decision making. |
| | LCA did broaden the perspective beyond CE's resource efficiency. However, it also introduced uncertainty and multiple impact categories, and thus complicated the efforts on CE strategies. | |

For the case company, the two circular strategies aligned with the parts of the product life cycle where insights were limited and uncertain. The lack of understanding of use, lifetime and EOL provided challenges in the inventory analysis of LCA, and thus also the applicability of the results for decision making. Throughout the case study, measures to increase insights into these aspects—such as dialog with customers and investigations of EOL practices and possibilities—were taken. The LCA results further lead the company to focus on increasing the lifetime of the case product through repair strategies, as well as efforts for informing and collaborating with customers (who perform re-test and re-furbishment). These measures provided a good fit with CBM development and the two circular strategies [18], but also with the phases in CBM development. The LCA activities throughout the case study were motivated by gaining knowledge on the environmental impacts through impact assessment results. However, the process itself turned out to be valuable for providing the case company with insight and knowledge on a part of the product's life cycle they traditionally have little knowledge on, insight that w important for CE efforts and CBM development. Table 2 shows how LCA can interact in CBM development in the different phases.

**Table 2.** The steps in CBM development and the possible role of LCA.

| Phases in CBM Development | The Potential Role of LCA in CBM Development |
| --- | --- |
| 1: Initiating phase | This case study confirmed literature findings [17,35–37] that it is difficult to fully utilize LCA on solutions that are under development, such as recycling technologies for EOL. However, LCA is a relevant tool to inform "what if" questions in initiating phases of CBM development. Further, LCAs foster collaboration and communication with stakeholders, which is essential for CBM development. |
| 2: Development phase | LCA can limit the risk of problem shifting by focusing on the entire product life cycle and multiple environmental categories, but it also brings complexity into CBM development through multiple impact categories. LCA can also be suitable for comparing different circular strategies, but this requires consistency in the assessment design. |
| 3: Implementing phase | LCA can be used to evaluate the performance of implemented circular strategies and shed light on feedback-effects. LCA as a part of an iterative CBM development process can be rewarding. However, social and economic sustainability are not evaluated by LCA and need to be examined by other measures. |

### 4.2. A Holistic Perspective on Circular Economy and Sustainable Development

The findings of this paper point to a more fundamental issue of how business concepts relate to the systemic issues inherent in sustainable development. This is a theoretical problem often perceived as a dilemma by scholars [45,46], i.e., the perceived need to focus levels of analysis either at the organizational or systemic level. The risk for problem shifting is a consequence, for example by neglecting important stakeholders or value dimensions. In such a paradoxical situation, the role of the researcher becomes essential—especially the way a phenomenon is interpreted and made accessible through analytical techniques. LCA is a tool for quantifying environmental impacts associated with a product or a service. It is a descriptive tool and therefore, in principle, neutral. The results themselves do not directly provide recommendations but merely state and document the environmental impacts. In contrast to LCA, CE is a strategic concept or framework based on a philosophy grounded in a set of values (e.g., a system without waste is the ideal state). Further, it is implicit that the shortest loops (Rs) are preferred over the longer loops (Rs) because they provide less strain on natural resources. CE is therefore more normative than LCA. As CBMs are ways of materializing CE, they are also normative. When CE and CBMs are the point of departure, a key assumption is already made, namely that closing the loop is the best path forward towards sustainable development. This case study exemplifies this interplay by the fact that

the LCA screening performed in 2012 showed that incineration with energy recovery was the most favorable option for waste treatment, when measuring environmental impacts. However, in further work with the case company, this was considered a less favorable option than other more circular alternatives. Incineration with energy recovery was not included as one of the scenarios to be explored in the innovation project that started in 2019, aiming at exploring EOL options for the case product. It was found through more normative reasoning that closing the loop through recycling and material recovery was preferable to energy recovery.

The results of an LCA point to the most important sections of a product's life cycle in terms of environmental pressure; this can both contribute to nudging efforts higher up in the waste hierarchy but can also point to lower parts as more environmentally beneficial. Thus, LCA as a descriptive tool is valuable in a more normative CE context.

The holistic focus of the impact assessment (including several environment impact categories) is also a way of avoiding problem shifting, i.e., creating or increasing another environmental problem when taking measures to decrease another. However, the same feature increases the complexity of the results and complicates the decision basis. In other words, the broad impact categories of LCA complicate the clear, 'simple' CE message of 'closing the loop'. Further, we warn about equating an evaluation by LCA, with ensuring the sustainability of CE efforts. LCA assesses only the environmental dimension of sustainability, although this is important, and so is the safeguarding of the social and economic sustainability. Especially social sustainability is in danger of being ignored or treated as implicit in CE efforts [6].

## 5. Conclusions

This study illustrates how LCA can act as an enabler for manufacturing companies that want to implement a circular business model, and the key lessons learned are as follows:

- Recognizing that the scope of LCA is usually product oriented, a CBM often has a wider, organizational scope;
- LCA can foster communication with stakeholders, which is valuable in CBM development;
- The use and EOL stages of a product life cycle are often the most difficult to find data on; the same stages are the core of the CE strategies, life-time extensions, and closed material loops;
- LCA as a quantitative tool can challenge CE's normative rule of closing the loop;
- LCA can confuse and complicate the "simplicity" of a CE strategy through multiple impact categories;
- Recognizing that LCA evaluates the environmental, not the social and economic sustainability of CBM.

The research methodology used to explore a phenomenon will influence the findings. An exploratory case study approach provides the opportunity for a deeper analysis but does not offer any generalizable conclusions on the use of LCA in CBM development; rather, it describes the experiences and discoveries from a single longitudinal case study.

For this case study, a B2B company was studied. An interesting focus point for further studies is what lessons and take-aways can be drawn on LCA and CBM development from B2C companies, and how the relationship between producer and end user affects these lessons. Moreover, the role of LCA in facilitating cooperation and commitment throughout the value chain to promote circularity should be further studied. Where the CE framework is systemic by nature, LCA is usually more product oriented. It can therefore be hypothesized that enabling CBM development by LCA, can be more achievable for companies with one product or a limited number of products. Further studies should investigate LCA as an enabler for CBM development for companies with more diverse product portfolios.

By providing an empirical example on how LCA and CBM development can interact, this paper is a contribution to the call for support in realizing the potential benefits of CE. By applying LCA, it also contributes to relating CE efforts to environmental sustainability. It does not, however, evaluate the social and economic dimensions. Further studies should explore this, ideally without increasing the complexity to a point where it becomes a barrier for implementation.

**Author Contributions:** Conceptualization, M.M.B. and S.S.V.; Formal analysis, M.M.B. and S.S.V.; Methodology, M.M.B. and S.S.V.; Writing—original draft, M.M.B. and S.S.V.; Writing—review and editing, M.M.B. and S.S.V. All authors have read and agreed to the published version of the manuscript.

**Funding:** This research was funded by the Research Council of Norway through SFI Manufacturing [grant number 237900] and IPN CompDetect [grant number 282018].

**Institutional Review Board Statement:** Not applicable.

**Informed Consent Statement:** Not applicable.

**Data Availability Statement:** Parts of the data presented in this study are contained within the study and found in the reference list, others are openly available. Some data are available on request from the corresponding author. Restrictions apply to the availability of parts of these data obtained from case company due to competitive reasons.

**Acknowledgments:** We would like to thank the case company, Hexagon Ragasco for their willingness to share experiences and Margrethe Skattum for valuable insights and reflections on this paper.

**Conflicts of Interest:** The authors declare no conflict of interest.

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
