# Peer review of "Life Cycle Assessment to Ensure Sustainability of Circular Business Models in Manufacturing"

_sustainability, doi:10.3390/su131911014_

Round 1

Reviewer 1 Report

The manuscript is well written and the research has been performed with accuracy. However, the manuscript should be improved in my view in particular in the discussion and in the conclusions. My suggestions are the following:

The discussion should be splitted in some subsections. I would avoid to discuss the results in the form of Table 1 and Table 2. In my view the Tables should only evidence some key aspects that you evidence in the discussion.  The discussed topics are good but their presentations should be much improved.

Morevoer the conclusions should be more coincise in order to appreciate the main results, the answering to the research questions, the research and practical implications, the limits of the study and proposals for future research.

Reviewer 2 Report

The authors analyze the applicability of Life Cycle Assessment as a path towards Circular Business Models (CBMs). They present a case study and discuss an example of a manufacturing company, focusing on two circular economy strategies, namely lifetime extensions and closing material loops.

The paper is well-structured and discusses an up-to-date research direction. The authors have performed a detailed study and made a broad review of current literature.

There are several minor issues that I suggest to be addressed before accepting the paper for publication:
1. The authors summarize the findings of their case study in Table 1. Hovewer, the discussion is lacking a clear summary how the applied methods contributed to extending the product lifetime or reducing material consumption. It would be good to see some numbers if the authors are allowed to share them.
2. The authors explain most of the used abbreviations, however, EOL (first mentioned in line 185) is not explained. I suppose it refers to "end of life", but it is not explicitly mentioned in the paper.
3. There are several language or typographic issues, e.g. "Manufacturing companies plays" (l. 39), "section3" (l. 151), "case studies enables" (l. 154).
